# Assessment of Popcorn’s Bioactive Status in Response to Popping

**DOI:** 10.3390/molecules29040807

**Published:** 2024-02-09

**Authors:** Jelena Vukadinović, Jelena Srdić, Natalija Kravić, Snežana Mladenović Drinić, Milena Simić, Milan Brankov, Vesna Dragičević

**Affiliations:** Maize Research Institute Zemun Polje, Slobodana Bajića 1, 11185 Belgrade, Serbia; jsrdic@mrizp.rs (J.S.); nkravic@mrizp.rs (N.K.); drinicsnezana2@gmail.com (S.M.D.); smilena@mrizp.rs (M.S.); mbrankov@mrizp.rs (M.B.); vdragicevic@mrizp.rs (V.D.)

**Keywords:** *β*-carotene, DPPH, functional food, GSH, tocopherols, *Zea mays* L. var. everta

## Abstract

Popcorn is a specialty maize variety with popping abilities. Although considered a snack, popcorn flakes provide a variety of benefits for the human diet. To evaluate the change in content of bioactive compounds in response to microwave popping, the kernels and flakes of twelve popcorn hybrids were assayed. Accordingly, the content of phytic acid, glutathione, phenolic compounds, carotenoids, and tocopherols, as well as the antioxidant activity, were evaluated. In all evaluated popcorn hybrids, the most pronounced significant average decrease of 71.94% was observed for GSH content, followed by 57.72% and 16.12% decreases for lutein + zeaxanthin and phytic acid content, respectively. In response to popping, in the majority of the evaluated hybrids, the most pronounced significant average changes of a 63.42% increase and a 27.61% decrease were observed for DPPH, followed by a 51.52% increase and a 24.48% decrease for *β*-carotene, as well as, a 48.62% increase and a 16.71% decrease for α-Tocopherol content, respectively. The applied principal component and hierarchical cluster analyses revealed the distinct separation of popcorn hybrids’ kernels and flakes, indicating the existence of a unique linkage of changes in bioactive compound content in response to popping.

## 1. Introduction

Maize with specific traits, such as popcorn (*Zea mays* L. everta) has snack status in many countries worldwide, and its use has increased in popularity over time. Popcorn flakes represent a wholegrain high nutrition value snack and provide a variety of benefits as food: they are low in calories, have a low glycaemic index, and are high in dietary fibre, protein, essential minerals, vitamins, and antioxidants, in higher concentrations than even some fruits and vegetables [1,2]. Likewise, popcorn is able to regulate excess stomach acidity and reduce feelings of hunger. Additionally, popcorn consumers have a satisfactory intake of fibre on a daily basis, approximately 22% higher than non-consumers [3]. Popcorn is considered a nutritious snack and a valuable food with excellent functional properties and thus could be considered an important source of phenolic compounds, especially ferulic acid [4]. Maize grain, taking in account various types (standard, sweet, popcorn) has the highest carotenoid content (lutein, zeaxanthin, *β*-carotene, and *β*-cryptoxanthin) of all cereals, especially in lutein and zeaxanthin which have a protective role in ocular tissue. Antioxidants such as tocopherols (*α*-, *β*-, *γ*-, *δ*-), glutathione (GSH), and phytic acid (Pphy), also found in all types of maize, have health-promoting roles, acting as antioxidants and protectors of biological membranes [5,6,7,8]. Although phytic acid acts as an antinutrient because it restrains the bioavailability of mineral elements in humans and non-ruminant animals [9], its benefits for health are numerous, from the anti-carcinogenic and anti-inflammatory, to a protective role against diabetes, cardio-vascular diseases, Alzheimer’s, and Parkinson’s disease [10].

The major difference between popcorn and other types of maize is its popping ability. The pericarp of popcorn maize is much harder and less porous, which allows the kernel to pop when the superheated water inside of it turns into steam and causes the hull to burst [11]. Generally, popcorn is popped with hot air, frying in hot oil, microwave heating or by gun puffing methods. To avoid the limitations of conventional popping (such as the rancidity of the oil) electromagnetic waves such as microwaves are used, allowing greater energy efficiency over a short period of time as well as the lower calorie values of flakes. Nowadays, under home conditions, popcorn is usually popped in microwaves oven due to its faster popping time and easier cleaning [12]. Moreover, many microwave ovens already have a popcorn programme installed. It is known that microwave popcorn paper bags contain smaller per- and polyfluoroalkyl substances (PFAS) “side-chains” which under certain conditions can detach and migrate to food from this use [13,14]. Such data indicate the need for replacing microwave paper bags with perhaps fireproof vessels or somehow reducing their use to obtain a higher quality and safer product. Some studies have been conducted on various popcorn genotypes regarding the popping properties of the kernels and their volume and shape expansion [15,16,17]. In other studies, the effect of maize cultivation, as well as popping technology (i.e., the addition of oil and the power of microwaves on the morphological properties of the flakes) have been estimated [18,19]. However, there is little information on the health-promoting compound content, such as antioxidants, in popcorn flakes. According to our knowledge, only a few studies have dealt with the effects of popping on antioxidant content [11,20,21,22]. It has been reported that the presence of oil could contribute to the increase in carotenoid level during the popping of yellow and purple popcorn kernels, while increasing the phenol content in flakes of yellow popcorn and decreasing it in flakes of purple popcorn [20,21]. Different findings regarding phenolic compound content have been reported. Reports on popping in a microwave have shown a non-significant effect on the total phenolic content in the popcorn kernel. However, by extracting phenolic compounds under the conditions of in vitro digestion, it was found that popcorn flakes have a higher antioxidant activity compared to the kernel [11]. Furthermore, it has been shown that increasing the power of infrared (IR) radiation, as a novel technique used for popping, resulted in higher total phenolic content in popcorn flakes [22].

Considering the growing demand for healthier snack products, i.e., popcorn flakes, more information is required regarding the effect of popping on the content of functional food ingredients. This study aimed at evaluating the effect of microwave popping in a fireproof vessel on the total phenolic (TPC), glutathione (GSH), phytic P (Pphy), carotenoid (i.e., lutein + zeaxanthin and *β*-carotene), and tocopherol (i.e., *α*-, *β*- + *γ*- and *δ*-) content, as well as the antioxidant activity, of twelve popcorn hybrids. The obtained results will point out to the health-beneficial value of popcorn flakes as a functional food.

## 2. Results and Discussion

The effect of popping was estimated by comparing the values of observed bioactive compounds between raw kernels and popcorn flakes, using two-way ANOVA. Based on the results, the analysed hybrids, popping treatment, and H–T interaction expressed significant impact on the content of the analysed antioxidants with the exception of TPC (Table 1). The highest coefficient of variation (4.77%) between the analysed factors was observed for GSH content, while the lowest (1.72%), was found for *δ*-Tocopherol content.

### 2.1. The Variability of Phytic P Acid, Glutathione, Total Phenolic Content, and DPPH Radical Scavenging Activity upon Popping

The content of Pphy, GSH, TPC, and DPPH radical scavenging activity in popcorn kernels and flakes after microwave popping in a fireproof vessel are presented in Table 2. The obtained values for TPC content in grain (1.49–2.49 mg FAE/g of DW) and flakes (1.56–2.56 mg FAE/g of DW) in this study were in agreement with the results reported by Bayomy [20]. For all twelve tested hybrids, the obtained data revealed that popping treatment led to a significant decrease in GSH content, as well as a non-significant increase in TPC (Table 2). In addition, for the majority of the evaluated hybrids, popping treatment generated a significant decrease in Pphy (Table 2). The popping treatment led to a significant decrease in DPPH radical scavenging activity in the vast majority of the evaluated hybrids, as well as to a significant increase in H1 and H2 (Table 2). The highest concentrations of Pphy, GSH, and TPC were found in the kernels of H6 (4.13 mg/g DW), H3 (2.69 µmol/g DW), and H10 (2.49 mg FAE/g DW), respectively. Similarly, the flakes of H7, H3, and H10 showed the highest content of Pphy (3.32 mg/g DW), GSH (0.74 µmol/g DW), and TPC (2.56 mg FAE/g DW), respectively. After popping treatment, the flakes of H6 (31.08%) showed the highest decrease, while H7 recorded the lowest decrease (0.86%), in Pphy content, when compared to the kernels. 

Such data indicate that the popping treatment has a positive impact on the nutritional quality of H6 flakes, from the standpoint of phytate antinutritive properties. Low-phytate grain provides potential benefits, especially in terms of higher bioavailability of minerals (iron, zinc, calcium, and magnesium) [7]. The flakes of H10 showed the largest reduction (80.61%), while the lowest reduction (63.61%) of GSH content was noticed in the flakes of H1 after popping, signifying it as an important genotype which retains, at the same time, a greater GSH pool after microwave popping and GSH source. This is an important trait, since it is well known that cereal grains are a low-GSH source in nutrition [23]. When compared to the kernels, the effect of popping led to an increased TPC content, being the most pronounced in flakes of H1 (4.27%) and the least pronounced in flakes of H12 (1.05%). A similar enhancement of total phenolic content in yellow popcorn flakes after popping treatment was reported [20,21]. By extracting phenolic compounds under the conditions of in vitro digestion, it was found that popcorn flakes have a higher antioxidant activity compared to the kernels [11]. The total phenolic content in popcorn flakes could be increased by increasing the power of infrared (IR) radiation, a novel technique used for popping [22]. Also, popping in a microwave oven with the addition of oil leads to an increased TPC content in popcorn flakes [21]. The observed increase in TPC in our study, irrespective of the fact that it is non-significant, supported the finding that the popping treatment in a microwave oven did not affect the content of phenolic compounds found in the kernels [9]. This increase in response to popping could be explained by the release of esterified and bound forms of phenolic acids from lignocellulose parts of damaged cell walls [24,25]. It is well known that DPPH radical scavenging activity greatly depends on genotype [26], which was supported by the results from this study. H4 kernels and H1 flakes had the highest DPPH radical scavenging activity. Using the FRAP method, Coco and Vinson [11] did not find significant differences regarding the antioxidant activity between popcorn kernels and flakes in response to microwave popping. However, they did determine a significant decrease in DPPH radical scavenging activity in flakes using IR power [22], which was in line with the results obtained from our study, i.e., that the popping significantly reduced DPPH radical scavenging activity in the flakes of most of the evaluated popcorn hybrids (Table 2).

### 2.2. The Variability of Carotenoid Content upon Popping

The content of carotenoids, i.e., lutein + zeaxanthin (L + Z) and *β*-carotene in popcorn kernels and flakes after microwave popping in a fireproof vessel is presented in Table 3. According to our best knowledge, the content of individual carotenoids upon popping is reported for the first time in this study. After microwave popping treatment, the content of lutein and zeaxanthin (L + Z) was decreased in the flakes of all analysed hybrids (Table 3). For *β*-carotene content, the trend after popping was slightly different. In response to popping treatment, the majority of the evaluated hybrids expressed a significant change in *β*-carotene content: six hybrids exhibited a decrease, while four hybrids exhibited an increase (Table 3). However, a non-significant increase in *β*-carotene content was observed in H5 and H12. The highest content of L + Z was found in the popcorn kernels of H7 (50.87 µg/g DW) and the flakes of H12 (28.86 µg/g DW). Similarly, the highest content of *β*-carotene was noticed in the popcorn kernels of H9 (1.98 µg/g DW) and the flakes of H7 (2.48 µg/g DW). After popping, the flakes of H7 (76.25%) showed the highest decrease; meanwhile, in the flakes of H12, the lowest decrease in L + Z content was noticed (19.19%), when compared to the kernels. According to the obtained data, the effect of popping led to the largest increase in *β*-carotene content in the flakes of H4 (92.07%), which was the opposite in the flakes of H11 (31.39%), in which the highest decrease was noticed, compared to the kernels. 

This data point out the positive effect of popping on *β*-carotene increase, particularly when hybrid H4 is considered, signifying it, together with H7 and H1, as a valuable source of *β*-carotene (antioxidant activity and as a precursor for vitamin A) from the nutritive and health-promoting points of view. The observed decrease in the total content of carotenoids (L + Z + *β*-carotene) in this study, after the microwave popping of twelve yellow popcorn hybrids, is in agreement with Pohndorf et al., Bayomy, and Paraginski et al. [16,20,21]. Undoubtedly, the reduction in carotenoid content can be related to the degradation thereof, particularly of lutein and zeaxanthin under heat exposure due to oxidation and isomerization reactions [20]. When looking at individual carotenoids, as in our study, it is important to emphasize that not all carotenoids decrease at the same time. A possible explanation for the existing trend in carotenoid alteration after microwave popping is the different intracellular localization of *β*-carotene and xanthophyll in the popcorn kernel tissue [27]. Furthermore, the content of *β*-carotene is a small share, and its increase does not lead to an increase in the total content of carotenoids, since lutein and zeaxanthin represented the majority of the carotenoids in popcorn kernels. Due to that, a false result interpretation can be created regarding the reduction in carotenoid content in popcorn flakes after popping treatments. Therefore, it is very important to determine the carotenoid profile in popcorn flakes because it gives a clearer picture of the effect of the popping treatment. The obtained result presents novelty, indicating that although there is a decrease in the total content of carotenoids in the flakes after microwave popping, the content of *β*-carotene increases, which additionally contributes to the higher nutritional value of popcorn flakes in terms of health-promoting compounds with low calorie values. Some studies reported that the addition of oil during popping leads to an increase in total carotenoid content in yellow popcorn flakes, compared to popping without oil addition [20,21]. This enhancement in total carotenoids could be explained by the fact that the soybean and maize oil used in the reported studies was already rich in naturally present carotenoids such as liposoluble antioxidants [28]. However, even though such popcorn flakes have a higher content of carotenoids, they also have a higher caloric value, which every consumer should be aware of. 

### 2.3. The Variability of Tocopherols Content upon Popping

The content of tocopherols, i.e., *α*-T, *β + γ*-T and *δ*-T, in popcorn kernels and flakes after microwave popping in a fireproof vessel is presented in Table 4. According to our best knowledge, this study reports, for the first time, the content of individual tocopherols upon popping. 

Generally, the data in the literature on the content of individual tocopherols upon heat effect (including microwaving) are scarce and, for that reason, it is very important to determine their content because they represent health-beneficial components that could contribute to popcorn’s value in terms of being a functional food. The obtained content range for *δ*-T (0.94–2.19 µg/g of DW), *β* + *γ*-T (12.89–19.70 µg/g of DW), and *α*-T (0.85–3.03 µg/g of DW) in popcorn kernels, was slightly lower compared to the results reported by Das and Singh [4]. The majority of the evaluated hybrids exhibited a significant decrease in *δ*-T and *β* + *γ*-T content in response to popping; however, a significant increase in *β* + *γ*-T content was observed in the hybrid H9 (Table 4). In response to microwave popping, the majority of the evaluated hybrids expressed significant change in *α*-T content: seven hybrids exhibited increase, while four of them exhibited decrease with non-significant decrease observed in H10 (Table 4). It was important to underline that the obtained data revealed a simultaneous increase in *α*-T and *β*-carotene content in popcorn flakes of H1, H2, H4, and H12 in response to microwave popping. The effect of microwave popping led to the largest decrease in *δ*-T content in flakes of H1 (53.98%), opposite to the H9 (0.81%) where the lowest decrease was noticed, regarding the kernel. After popping treatment, the highest reduction in *β* + *γ*-T content was noticed in flakes of H1 (50.18%), while in flakes of H9 (35.46%) the greatest increase in *β* + *γ*-T content was achieved, when compared to the kernels. The effect of popping led to the largest increase in *α*-T content in the flakes of H9 (167.23%), which was the opposite effect to the flakes of H8 (29.56%), in which the highest decrease was noticed, compared to the kernels. The observed increase in *α*-T content in popcorn flakes in our study could be explained by the thermal stability of tocopherols (up to 200 °C) [29]. It has been shown that short heating treatments promote an increase in *α*-T content while longer heating treatments result in a reduction in *α*-T content [30], supporting the data obtained in our study, because popcorn flakes mostly had a higher *α*-T content after only 5 min of microwave popping. Such data highlight microwave popping as a tool to enhance the health-promoting compounds in popcorn flakes. Similarly, as in this study, Bernhardt and Schlich [31] reported higher *α*-T content after the microwave preparation of frozen broccoli.

### 2.4. The PCA of the Evaluated Bioactive Compounds

In order to assess the possible relationship between popcorn hybrids, microwave popping, and the analysed bioactive compounds, the PCA was applied. PCA resulted in informative the four-component model (82.43% of the overall data variance) and mutual projections for these PCs (factor scores and loadings) are shown in Figure 1. The four-component PCA model resulted in eigenvalues greater than 1, supporting the validity and significance of the presented data (Appendix A). The first two PCs contributed the most to accurate separation and grouping of the observed popcorn kernels and flakes (Figure 1a). It was found that the content of *α*-T and *β*-carotene were the most efficient parameters for distinguishing the unique antioxidant profiles of the evaluated popcorn kernels and flakes (Figure 1b); pointing out the importance of these bioactive compounds for popcorn flakes’ response to popping. Within each group (either the group of popcorn kernels or popcorn flakes), two subgroups could be observed. Within the popcorn kernels group, the most efficient parameters for separation of H1, H2, H7, H9, and H10 were *δ*-T and Pphy content (lower right quadrant, Figure 1a,b). In addition, the content of TPC, DPPH, GSH, L + Z, and *β + γ*-T had the highest influence on the separation of H3, H4, H5, H6, H8, H11, and H12 (upper right quadrant, Figure 1a). This grouping was totally confirmed by applied agglomerative hierarchical cluster analysis (HCA) at the level of similarity of 6.5% (Figure 2). Within the popcorn flakes group, the allocation of H3, H5, H8, H11, and H12 (upper left quadrant, Figure 1a,b) was highly influenced by the *α*-T content.

In addition, the β-carotene content led to the distinction of H1, H2, H4, H6, H7, H9, and H10. Formation of these two subgroups of the evaluated hybrids was confirmed by the HCA at the level of similarity of 6.5% (Figure 2). The obtained results were in line with a similar trend reported in the change in *β*-carotene and *α*-tocopherol content upon heat treatment [31]. Despite the great variability in bioactive compound content among the evaluated popcorn hybrids, the hybrids’ unique responses to popping were observed.

## 3. Material and Methods

### 3.1. Field Trial and Sample Preparation

The experiments were conducted in 2020 at the Maize Research Institute “Zemun Polje”, Belgrade, Serbia (44°52′ N, 20°19′ E, 81 m a.s.l.), on a slightly calcareous chernozem, with 53.0% sand, 30.0% silt, 17.0% clay, 3.8% organic matter, 7.0 pH KCl, and 7.17 pH H_2_O. In this study, twelve yellow popcorn hybrids were sown in April and the elementary plot encompassed 7 m^2^ (40 plants), in a completely randomized block design in 4 replications. Harvest was performed in September and during vegetation all standard cropping practices were applied. After harvest, kernel moisture was brought to 14% and, after that, the kernels were milled on a Perten 120—Sweden (particle size < 500 μm), as were the flakes after popping treatment. 

The popping treatment was performed in a microwave oven (Samsung, Seoul, Republic of Korea, model MS23F301) at 700 W for 5 min on the Instant program (the manufacturer’s recommendation for making popcorn). For the popping treatment, 1/2 a cup of yellow popcorn kernels (approximately 100 g of popcorn kernels), was poured into a fireproof vessel and was covered with a ceramic plate. Neither oil, salt, nor sugar were added to the popcorn kernels. The obtained and milled samples were stored at −21 °C and analyses were conducted within 48 h. The content of all analysed phytochemical compounds from the popcorn kernels and flakes was calculated on the dry weight (DW), which was obtained after drying in a ventilation oven (EU instruments, EUGE425, Novo Mesto, Slovenia, EU) to constant weight (105 °C for 4 h).

### 3.2. Chemicals and Reagents

Standards of the carotenoids (lutein, zeaxanthin, *β*-carotene), tocopherols (*α*-T, *γ*-T, *δ*-T), as well as the ferulic acid, acetone, ethanol, methanol, ethyl acetate, 2-propanol, all HPLC grade, were obtained from Sigma-Aldrich (Munich, Germany). From the same supplier Folin–Ciocalteu reagent, sodium acetate, sodium carbonate, and TCA (trichloroacetic acid) were also purchased. TRIS (2-amino-2-hydroxymethyl-propane-1,3-diol), DTNB (2,2′-dinitro-5,5′-dithiobenzoic acid), iron (III) chloride hexahydrate, DPPH^•^ (2,2-diphenyl-1-picrylhydrazyl), and 5′-sulphoosalicylic acid were purchased from Merck (Darmstadt, Germany). Ultrapure water, conductivity 0.055 μS/cm (TKA, Dreieich, Germany), was used to prepare the appropriate standard solutions and samples, while Target2^®^ syringe filters (PTF membrane, 0.45 μm, diameter 17 mm) were used to filter the extracts.

### 3.3. Spectrophotometric Methods

#### 3.3.1. Phytic P Content (Pphy)

Phytic acid determination was performed using the method of Dragičević et al. [32] with some modification. Extraction was accomplished by using 10 mL 5% TCA solution on the popcorn kernels or flakes (approximately 0.5 g). Extracts were homogenized on a vertical shaker (30 min at room temperature) and, after centrifugation (12,000 rpm for 15 min (Dynamica—Model Velocity 18R Versatile Centrifuge, Rotor TA15-24-2) at 4 °C), the obtained clear supernatant was used for both phytic acid and total glutathione content. An aliquot (0.25 mL) was mixed with distilled water (1.25 mL) and 0.5 mL of Wade reagent (0.3 g of FeCl_3_ × 6H_2_O + 3 g of 5′-sulphosalicylic acid in one litre). After developing the pink colour, absorbance was measured using a spectrophotometer (Biochrom Libra S22 UV/Vis Spectrophotometer—Biochrom, Cambridge, UK) at λ = 500 nm against a blank (containing 5% TCA instead of a sample). The Pphy content was reported as the mean value of the four replications and expressed as mg/g of DW sample.

#### 3.3.2. Total Glutathione Content (GSH)

Total glutathione was determined according to Sari Gorla et al. [33]. An aliquot of the appropriate supernatant (0.5 mL) was mixed with 0.5 mL of distilled water. Then, 1 mL of reagent containing 0.1 M TRIS (pH = 7) and 1.5 mM DTNB was added. After centrifugation of the mixture, the absorbance was read at 412 nm against a blank that instead of the sample contained 0.5 mL of distilled water, i.e., 5% TCA. The total glutathione content was calculated based on the measured absorbance values of the samples and the molar absorption coefficient of 1.36 × 104 dm^3^ mol^−1^ cm^−1^ at 412 nm [34]. The content was expressed as µmol/g of DW sample and reported as the mean value of four replications.

#### 3.3.3. Total Phenolic Content (TPC)

Determination of TPC in popcorn kernels or flakes was performed according to the procedure described by Singleton et al. [35]. Approximately 0.3 g of sample was extracted with 10 mL of an acetone–water mixture (7:3, *v*/*v*) on a vertical shaker for 30 min. After centrifugation, the obtained clear supernatant was used for both the total phenolic content and the antioxidant activity. An aliquot of the supernatant (0.2 mL) was mixed with 0.5 mL of distilled water. After adding 0.25 mL of Folin–Ciocalteu reagent (2 M relative to the acid) and 1.25 mL of Na_2_CO_3_ (20%), the cuvettes were vigorously shaken and left in the dark for 40 min. After that, the absorbance of the blue-coloured mixture was measured at 722 nm against a blank probe (containing acetone instead of a sample). The total phenolic content was reported as the mean value of four replications and expressed as mg of ferulic acid equivalents (FAE) per g of DW sample.

#### 3.3.4. Antioxidant Activity—DPPH^•^ Method

DPPH radical scavenging activity was determined according to the method suggested by Abe et al. [36]. An aliquot of the supernatant (0.3 mL) was mixed with 0.2 mL of an acetone–water mixture (7:3, *v*/*v*), 0.25 mL of DPPH^•^ reagent (0.5 mM), and 0.5 mL of acetate buffer (100 mM, pH = 5). After standing the mixture for 30 min in the dark, the absorbance was measured at 517 nm against a blank sample that, instead of the sample, contained 0.3 mL of absolute ethanol. The radical scavenging activity (*RSA*) of the sample reported as the mean value of three measurements and was calculated as the percentage of reacted DPPH^•^ reagent, using Equation (1):(1)$$RSA(\%)=\frac{\left(A_{DPPH}-A_{sample}\right)}{A_{DPPH}}\times 100$$
where A_DPPH_ is the absorbance of a DPPH^•^ reagent (blank), while A_sample_ is the absorbance of popcorn kernels or flakes.

### 3.4. HPLC Analysis

For the determination of tocopherol and carotenoid content, approximately 0.2 g of popcorn kernels and flakes was used. The extraction of carotenoids (lutein + zeaxanthin (L + Z) and *β*-carotene) was achieved by adding a mixture of methanol and ethyl acetate (6:4, *v*/*v*), 2 × 5 mL, while for tocopherol extraction (*α*-T, *β + γ*-T and *δ*-T) 5 mL of 2-propanol was added [37]. After homogenization in an ultrasound bath (30 min at 25 °C) for all analyses, the extracts were centrifuged (3000 rpm for 5 min), filtered through the syringe filter, and directly injected into the HPLC system (Dionex UltiMate 3000 Thermo Scientific, Bremen, Germany). Only carotenoid extracts before injection were firstly evaporated to dryness under a stream of nitrogen and then redissolved in the 1 mL of the mobile phase. The same analytical column (Acclaim Polar Advantage II, C18 (150 × 4.6 mm, 3 μm)) and mobile phase for separation and detection of carotenoids and tocopherols, as well as the chromeleon software package (version 7.2) for instrument control, data acquisition, and analysis, were the same as described in study by Mesarovic et al. [38]. The detection of tocopherols and carotenoids was accomplished by fluorescence (λ_ex_ 290 nm; λ_em_ 325 nm) and a photodiode array (at 450 nm and 470 nm) detector, respectively. The analysis of carotenoids lasted for 20 min, with a thermostated column at 25 °C and the injection volume of the extract volume was 100 µL. Tocopherol chromatographic separation lasted for 15 min on a column thermostated at 30 °C, while the injection volume of the extract was 5 µL. Tocopherol and carotenoid content is reported as the mean value of three independent injections and is expressed as µg/g of DW.

### 3.5. Statistical Analysis

To analyse the data on the evaluated phytochemical parameters, two-factorial analysis of variance (ANOVA) set up according to the randomized complete block design (RCBD) was applied using the SPSS software for Windows, version 14.0 (SPSS Inc., Chicago, IL, USA). To determine the differences between the hybrids (H) and the popping treatment (T), as well as the hybrid–popping treatment interaction (H × T), Fisher’s least significant difference (LSD) test at a 0.95 confidence level (*p* ≤ 0.05) was conducted. Furthermore, to determine the relationship between the content of the evaluated phytochemical parameters in the popcorn kernels and flakes in response to microwave popping, principal component analysis (PCA) and agglomerative hierarchical cluster analysis (HCA) were carried out using MATLAB (R2011a) with the PLS Toolbox software package (v.6.2.1). For analysis, the obtained data were mean-centred, auto-scaled, and the singular value decomposition (SVD) algorithm was employed (95% confidence level) for Hotelling T2 limits. A percent change (% change) was used to determine the trend of the evaluated phytochemical content in response to popping treatment.

## 4. Conclusions

This study provides a comprehensive understanding of the advantages and disadvantages of microwave popping in a fireproof vessel on the bioactive compound content (i.e., phytic acid, glutathione, phenolic compounds, lutein + zeaxanthin, *β*-carotene, *δ*-, *β* + *γ*-, and *α*-tocopherols) of twelve yellow popcorn hybrids. While microwave popping treatment led to a significant decrease in Pphy, GSH, and L + Z content, no significant change in the TPC content was noted in any of the evaluated popcorn hybrids. The application of PCA and HCA confirmed the existence of unique linkages between changes in the bioactive compound content in response to popping in a hybrid-dependent manner, emphasizing the positive effect of microwave popping in terms of increasing the content of the α-tocopherol and β-carotene found in H1, H2, H4, and H7. Today, when demands for a healthier lifestyle and the identification of health-beneficial components are in focus, the results of this study may lead to new opportunities for the breeding of popcorn hybrids with higher bioactive compound contents, i.e., for their promotion as a source of functional food. 

## Figures and Tables

**Figure 1 molecules-29-00807-f001:**
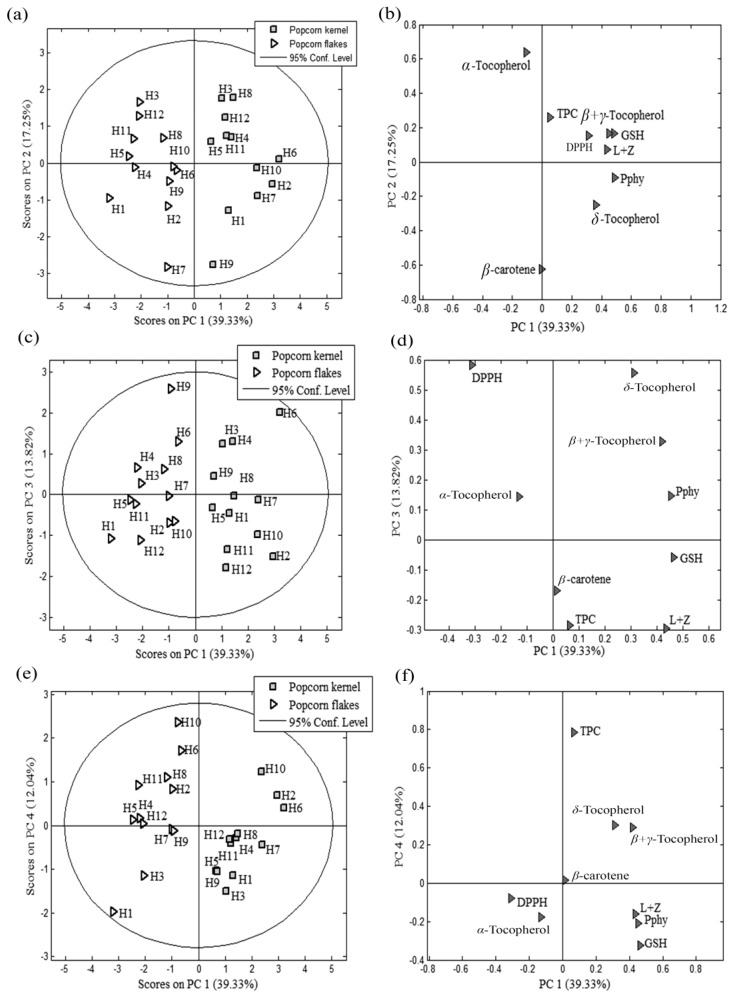
The applied PCA for the evaluated bioactive compound content in popcorn kernels and flakes upon popping: scores (**a**,**c**,**e**) and loading plot (**b**,**d**,**f**).

**Figure 2 molecules-29-00807-f002:**
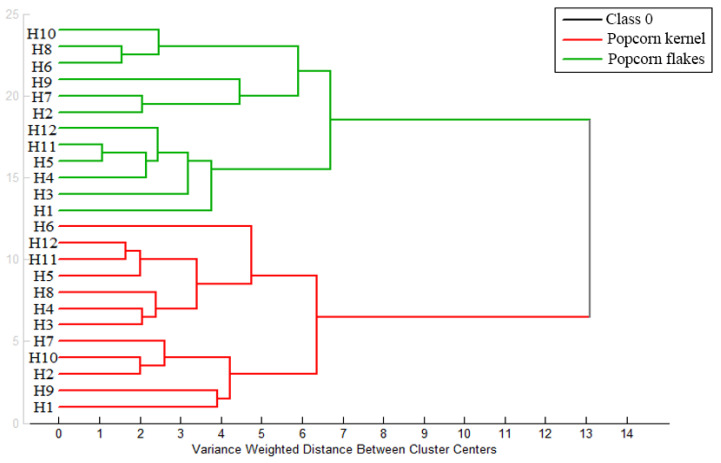
The applied agglomerative hierarchical cluster analysis for the evaluated bioactive compound content in popcorn kernels and flakes upon popping.

**Table 1 molecules-29-00807-t001:** The ANOVA and LSD values of the evaluated bioactive compounds. Abbreviations: H—popcorn hybrid; T—popping treatment; H × T —hybrid by popping treatment interaction; CV—coefficient of variation; Pphy—phytic acid; GSH—total glutathione content; TPC—total phenolics content; DPPH—radical scavenging activity; *δ*-T–, *β* + *γ*-T–, *α*-T—Tocopherols; L + Z—Lutein + Zeaxanthin.

	Mean Square			LSD		
Antioxidants	H	T	H × T	H	H × T	CV (%)
Pphy	0.12 *	3.83 *	0.11 *	0.13	0.18	2.89
GSH	113,810.89 *	31,040,093.47 *	70,256.24 *	99.19	140.30	4.77
TPC	38.17 *	2.45	0.02	0.14	0.19	4.46
DPPH	334.63 *	1383.72 *	439.16 *	2.90	4.10	3.15
*δ*-T	0.65 *	0.91 *	0.04 *	0.05	0.07	1.72
*β* + *γ*-T	8.67 *	91.30 *	9.45 *	0.41	0.58	1.80
*α*-T	1.14 *	0.40 *	0.43 *	0.05	0.07	1.90
L + Z	86.18 *	5666.82 *	92.29 *	0.83	1.18	2.23
*β*-carotene	0.66 *	0.03 *	0.23 *	0.05	0.07	2.62

* significant at a 0.05 probability level; LSD—Fisher’s least significant difference test at a 0.95 confidence level.

**Table 2 molecules-29-00807-t002:** The content * of Pphy, GSH, TPC, and DPPH radical scavenging activity in the kernels and flakes of twelve popcorn hybrids after microwave popping.

Hybrid	Pphy (mg/g DW)			GSH (µmol/g DW)			TPC (mg FAE/g DW)			DPPH (%)		
	Kernels	Flakes	% Change	Kernels	Flakes	% Change	Kernels	Flakes	% Change	Kernels	Flakes	% Change
H1	3.30 ± 0.08 ^ef^	2.72 ± 0.04 ^ijk^	−17.54	1.83 ± 0.01 ^f^	0.66 ± 0.00 ^gh^	−63.61	1.49 ± 0.03 ^k^	1.56 ± 0.02 ^jk^	4.27	43.89 ± 0.34 ^l^	79.13 ± 0.13 ^bc^	80.27
H2	3.55 ± 0.14 ^b^	3.24 ± 0.02 ^ef^	−8.67	1.92 ± 0.14 ^f^	0.59 ± 0.00 ^hi^	−69.32	2.30 ± 0.03 ^c^	2.34 ± 0.05 ^bc^	1.77	51.14 ± 0.55 ^k^	74.97 ± 0.05 ^de^	46.60
H3	3.36 ± 0.09 ^bcdef^	2.88 ± 0.04 ^hi^	−14.35	2.69 ± 0.00 ^a^	0.74 ± 0.01 ^g^	−72.49	1.74 ± 0.04 ^hij^	1.76 ± 0.04 ^ghi^	1.68	81.41 ± 0.05 ^bc^	57.87 ± 0.16 ^j^	−28.92
H4	3.49 ± 0.05 ^bcd^	2.76 ± 0.15 ^ijk^	−20.89	2.12 ± 0.05 ^e^	0.65 ± 0.08 ^ghi^	−69.43	2.03 ± 0.16 ^ef^	2.08 ± 0.07 ^def^	2.17	85.88 ± 0.08 ^a^	69.79 ± 0.05 ^fg^	−18.74
H5	3.51 ± 0.02 ^bc^	2.63 ± 0.13 ^k^	−25.00	2.16 ± 0.04 ^de^	0.60 ± 0.03 ^ghi^	−72.19	2.04 ± 0.20 ^ef^	2.09 ± 0.04 ^def^	2.30	70.40 ± 0.27 ^fg^	68.10 ± 0.18 ^gh^	−3.28
H6	4.13 ± 0.02 ^a^	2.84 ± 0.09 ^ij^	−31.08	2.68 ± 0.00 ^a^	0.68 ± 0.03 ^gh^	−74.60	2.24 ± 0.03 ^cd^	2.29 ± 0.05 ^c^	2.46	73.12 ± 0.18 ^ef^	57.42 ± 0.23 ^j^	−21.48
H7	3.35 ± 0.02 ^cdef^	3.32 ± 0.02 ^def^	−0.86	2.10 ± 0.07 ^e^	0.63 ± 0.06 ^ghi^	−69.91	1.89 ± 0.04 ^fgh^	1.94 ± 0.05 ^fg^	2.61	80.79 ± 0.12 ^bc^	60.84 ± 0.25 ^ij^	−24.70
H8	3.22 ± 0.07 ^fg^	2.86 ± 0.06 ^hi^	−11.04	2.31 ± 0.09 ^c^	0.60 ± 0.02 ^ghi^	−73.86	2.21 ± 0.10 ^cde^	2.27 ± 0.04 ^cd^	2.38	77.41 ± 0.09 ^cd^	64.27 ± 0.10 ^hi^	−16.97
H9	3.30 ± 0.08 ^ef^	3.04 ± 0.03 ^gh^	−7.72	1.84 ± 0.22 ^f^	0.52 ± 0.05 ^ij^	−71.69	1.67 ± 0.16 ^ijk^	1.70 ± 0.07 ^ij^	1.92	81.59 ± 0.38 ^ab^	46.67 ± 0.10 ^l^	−42.80
H10	3.20 ± 0.01 ^fg^	2.71 ± 0.29 ^ijk^	−15.09	2.27 ± 0.06 ^cd^	0.44 ± 0.00 ^j^	−80.61	2.49 ± 0.08 ^ab^	2.56 ± 0.15 ^a^	2.82	56.65 ± 0.68 ^j^	32.38 ± 0.04 ^m^	−42.84
H11	3.40 ± 0.02 ^bcde^	2.61 ± 0.04 ^k^	−23.35	2.38 ± 0.02 ^bc^	0.60 ± 0.0 ^hi^	−74.88	2.35 ± 0.19 ^bc^	2.38 ± 0.08 ^abc^	1.45	70.91 ± 0.16 ^efg^	44.33 ± 0.07 ^l^	−37.48
H12	3.25 ± 0.05 ^ef^	2.67 ± 0.04 ^jk^	−17.78	2.45 ± 0.05 ^b^	0.72 ± 0.02 ^gh^	−70.66	2.30 ± 0.02 ^bc^	2.32 ± 0.06 ^bc^	1.05	52.89 ± 0.12 ^k^	45.18 ± 0.12 ^l^	−14.58
Average	3.42 ± 0.25	2.86 ± 0.23	−16.12	2.23 ± 0.29	0.62 ± 0.08	−71.94	2.06 ± 0.31	2.11 ± 0.31	2.24	68.84 ± 14.11	58.41 ± 14.00	63.44
												−25.18

* Contents are expressed as mean values (n = 3) ± standard deviation (SD); means followed by a different letter within the same row are significantly different based on Fisher’s least significant difference test at a α = 0.05 level.

**Table 3 molecules-29-00807-t003:** The carotenoid content * (µg/g DW) in the kernels and flakes of twelve popcorn hybrids after microwave popping.

Hybrid	Lutein + Zeaxanthin			*β*-Carotene		
	Kernels	Flakes	% Change	Kernels	Flakes	% Change
H1	24.90 ± 0.43 ^i^	13.43 ± 0.04 ^k^	−46.04	1.03 ± 0.02 ^i^	1.26 ± 0.03 ^g^	22.11
H2	45.14 ± 0.67 ^b^	13.24 ± 0.04 ^kl^	−70.67	1.74 ± 0.03 ^c^	2.05 ± 0.06 ^b^	17.68
H3	26.28 ± 0.62 ^h^	14.22 ± 0.13 ^k^	−45.91	0.72 ± 0.03 ^l^	0.51 ± 0.01 ^m^	−29.21
H4	32.58 ± 1.62 ^f^	11.15 ± 0.01 ^m^	−65.77	0.85 ± 0.05 ^k^	1.64 ± 0.04 ^d^	92.07
H5	31.54 ± 1.73 ^f^	12.14 ± 0.03 ^lm^	−61.51	1.14 ± 0.07 ^h^	1.14 ± 0.03 ^h^	0.68
H6	28.06 ± 0.15 ^g^	14.22 ± 0.07 ^jk^	−49.32	1.05 ± 0.01 ^i^	0.87 ± 0.02 ^k^	−16.38
H7	50.87 ± 0.61 ^a^	12.08 ± 0.03 ^lm^	−76.25	1.42 ± 0.02 ^ef^	2.48 ± 0.07 ^a^	74.23
H8	40.57 ± 0.74 ^d^	14.12 ± 0.18 ^k^	−65.19	1.02 ± 0.02 ^ij^	0.82 ± 0.02 ^k^	−19.45
H9	34.72 ± 0.06 ^e^	14.27 ± 0.04 ^jk^	−58.91	1.98 ± 0.01 ^b^	1.57 ± 0.04 ^d^	−20.67
H10	43.24 ± 0.23 ^c^	13.35 ± 0.03 ^k^	−69.12	1.36 ± 0.01 ^f^	0.96 ± 0.02 ^j^	−29.76
H11	43.64 ± 0.12 ^c^	15.40 ± 0.05 ^j^	−64.72	1.46 ± 0.04 ^e^	1.00 ± 0.02 ^ij^	−31.39
H12	35.72 ± 0.09 ^e^	28.86 ± 0.27 ^g^	−19.19	1.15 ± 0.05 ^h^	1.20 ± 0.03 ^gh^	4.67
Average	36.44 ± 8.25	14.71 ± 4.61	−57.72	1.24 ± 0.37	1.29 ± 0.56	35.24
						−24.48

* Contents are expressed as mean values (*n* = 3) ± standard deviation (SD); means followed by a different letter within the same row are significantly different based on Fisher’s least significant difference test at a α = 0.05 level.

**Table 4 molecules-29-00807-t004:** The tocopherol content * (µg/g DW) in the kernels and flakes of twelve popcorn hybrids after microwave popping.

Hybrid	*δ*-T			*β + γ*-T			*α*-T		
	Kernels	Flakes	% Change	Kernels	Flakes	% Change	Kernels	Flakes	% Change
H1	1.33 ± 0.01 ^j^	0.61 ± 0.01 ^p^	−53.98	16.94 ± 0.15 ^ef^	8.44 ± 0.48 ^l^	−50.18	0.85 ± 0.02 ^p^	1.74 ± 0.03 ^k^	105.01
H2	1.46 ± 0.01 ^gh^	1.09 ± 0.02 ^l^	−25.13	19.07 ± 0.13 ^b^	14.45 ± 0.18 ^i^	−24.19	1.64 ± 0.04 ^l^	1.94 ± 0.02 ^j^	17.75
H3	1.35 ± 0.03 ^ij^	0.96 ± 0.02 ^mn^	−28.61	18.13 ± 0.25 ^c^	13.32 ± 0.26 ^jk^	−26.54	2.92 ± 0.07 ^b^	3.03 ± 0.06 ^a^	3.97
H4	1.70 ± 0.01 ^e^	1.20 ± 0.02 ^k^	−29.05	17.77 ± 0.12 ^cd^	14.26 ± 0.20 ^i^	−19.76	2.14 ± 0.01 ^h^	2.58 ± 0.04 ^e^	20.83
H5	0.99 ± 0.01 ^m^	0.91 ± 0.01 ^n^	−8.22	15.25 ± 0.19 ^h^	13.40 ± 0.19 ^j^	−12.15	2.20 ± 0.03 ^h^	2.07 ± 0.03 ^i^	−5.75
H6	2.19 ± 0.02 ^a^	1.86 ± 0.03 ^c^	−15.03	19.70 ± 0.17 ^a^	16.85 ± 0.24 ^f^	−14.49	1.89 ± 0.03 ^j^	1.51 ± 0.01 ^m^	−20.28
H7	1.74 ± 0.05 ^de^	1.40 ± 0.02 ^hi^	−19.15	17.61 ± 0.52 ^cd^	13.17 ± 0.21 ^jk^	−25.21	1.33 ± 0.02 ^n^	1.48 ± 0.04 ^m^	10.98
H8	1.51 ± 0.04 ^fg^	1.40 ± 0.02 ^hi^	−7.23	17.29 ± 0.26 ^def^	16.09 ± 0.22 ^g^	−6.92	3.05 ± 0.04 ^a^	2.15 ± 0.05 ^h^	−29.56
H9	1.95 ± 0.04 ^b^	1.93 ± 0.02 ^b^	−0.81	12.89 ± 0.15 ^jk^	17.46 ± 0.16 ^de^	35.46	1.01 ± 0.01 ^o^	2.69 ± 0.05 ^d^	167.23
H10	1.78 ± 0.03 ^d^	1.56 ± 0.01 ^f^	−12.41	17.30 ± 0.18 ^def^	15.44 ± 0.12 ^h^	−10.75	1.55 ± 0.01 ^m^	1.49 ± 0.03 ^m^	−3.72
H11	0.95 ± 0.02 ^mn^	0.93 ± 0.01 ^mn^	−2.38	15.32 ± 0.29 ^h^	13.99 ± 0.19 ^i^	−8.68	2.32 ± 0.04 ^g^	2.06 ± 0.05 ^i^	−11.23
H12	0.94 ± 0.06 ^n^	0.69 ± 0.01 ^o^	−26.93	15.46 ± 0.73 ^h^	12.75 ± 0.16 ^k^	−17.51	2.46 ± 0.03 ^f^	2.82 ± 0.07 ^c^	14.58
Average	1.49 ± 0.40	1.21 ± 0.43	−53.98	16.89 ± 1.88	14.13 ± 2.35	35.46	1.95 ± 0.70	2.13 ± 0.54	48.62
						−19.67			−14.11

* Contents are expressed as mean values (*n* = 3) ± standard deviation (SD); means followed by a different letter within the same row are significantly different based on Fisher’s least significant difference test at a α = 0.05 level.

## Data Availability

The data presented in this study are available in the article.

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
