# Peer review of "Assessment of Popcorn’s Bioactive Status in Response to Popping"

_molecules, 2024, doi:10.3390/molecules29040807_

Round 1

Reviewer 1 Report

Comments and Suggestions for Authors

1) Sentence in lines 9-11 in Abstract is unclear, consider rephrasing the sentence. 

2) On lines 66-67, authors included this statement- "On the other hand, popping popcorn in a microwave oven does not significantly affect the total phenolic content in the popcorn kernel [9]". Authors need to further elaborate the difference of the cited study with the current study. 

3) Equation 1- Adpph should be corrected to ADPPH

4) Table 1- data for mean square, specifically for DPPH should have similar decimal points like the other data (2 decimal point is sufficient).

5) Table 2- the increase in TPC following the popping treatment was not significant in all 12 hybrids. Authors need to rephrase the statement on line 207 to further clarify this. The same for the decrease in DPPH for hybrid 5, explained in lines 207-209.

6) Line 226-229- the increase of TPC discussed here is irrelevant with the non-significant differences detected between the kernel and the flakes. In this regard, the discussion interpreting the results on impact of popping to TPC stated on lines 231-244 needs to be revised. 

7) The decrease of β-carotene in H5 and H12 was also not significant, thus elaboration of results on lines 261-263 needs to be rephrased. 

8) Line 288: remove "are"

9) The decrease of δ-T in H9, H11 as explained in lines 317-319 is inaccurate, since the decrease were not significant. Authors need to rephrase this data interpretation. This is the same for the explanation on the decrease of α-T in H10, on line 322. 

10) In relation to comments number 4-6 & 8, all these points should also be considered by the authors to rephrase the summary of findings included in the conclusion section (lines 384-385). 

11) Only about 25% of the references are current (2018 onwards). Authors are encouraged to highlight more from current literatures, to help emphasize the relevance of the study.

12) Line 347 - "Fig. 1b" could be written as "Figure 1b" (as seen in the other part of this manuscript for other figures). 

Author Response

Comments and Suggestions for Authors

  1. Sentence in lines 9-11 in Abstract is unclear, consider rephrasing the sentence.
  • According to the Reviewer’s comment, among other changes, the two first sentences in the Abstract were revised.

  1. On lines 66-67, authors included this statement- "On the other hand, popping popcorn in a microwave oven does not significantly affect the total phenolic content in the popcorn kernel [9]". Authors need to further elaborate the difference of the cited study with the current study.
  • According to the Reviewer’s comment, the sentence was revised (due to the changes in the manuscript body text, the cited study is now Reference no. 11).

  1. Equation 1- Adpph should be corrected to ADPPH.
  • Accepted and corrected.

  1. Table 1- data for mean square, specifically for DPPH should have similar decimal points like the other data (2 decimal point is sufficient).
  • Accepted and corrected.

  1. Table 2- the increase in TPC following the popping treatment was not significant in all 12 hybrids. Authors need to rephrase the statement on line 207 to further clarify this. The same for the decrease in DPPH for hybrid 5, explained in lines 207-209.
  • According to the Reviewer’s suggestion, the sentence was revised.

  1. Line 226-229- the increase of TPC discussed here is irrelevant with the non-significant differences detected between the kernel and the flakes. In this regard, the discussion interpreting the results on impact of popping to TPC stated on lines 231-244 needs to be revised.
  • According to the Reviewer’s comment, a part of the discussion interpreting the results on popping impact to TPC was revised.

  1. The decrease of β-carotene in H5 and H12 was also not significant, thus elaboration of results on lines 261-263 needs to be rephrased.
  • According to the Reviewer’s comments, the elaboration of results regarding β-carotene was rephrased.

  1. Line 288: remove "are"
  • Accepted and corrected.

  1. The decrease of δ-T in H9, H11 as explained in lines 317-319 is inaccurate, since the decrease were not significant. Authors need to rephrase this data interpretation. This is the same for the explanation on the decrease of α-T in H10, on line 322.
  • According to the Reviewer’s comment, the elaboration of results regarding tocopherols was corrected.

  1. In relation to comments number 4-6 & 8, all these points should also be considered by the authors to rephrase the summary of findings included in the conclusion section (lines 384-385).
  • According to the Reviewer’s comment, the Conclusions Section was revised.

  1. Only about 25% of the references are current (2018 onwards). Authors are encouraged to highlight more from current literatures, to help emphasize the relevance of the study.
  • According to the Reviewer’s suggestion, the more up-to-date literature was provided.

  1. Line 347 - "Fig. 1b" could be written as "Figure 1b" (as seen in the other part of this manuscript for other figures).
  • Accepted and corrected.

Reviewer 2 Report

Comments and Suggestions for Authors

The authors should omit any discussion of analyte changes during the microwave process that are not significant.

The use of "flakes" is not common to describe popped popcorn.  More appropriate is "popcorn" or "kernels"

line 53  per- not per

line 113 no standard is mentioned.

On each table at the bottom of the data the authors should put the average of all the 12 hybrids and the standard deviation and the statistics for the average % changes due to the microwave process.

Comments on the Quality of English Language

English is very well done with few grammar changes needed.

Author Response

Comments and Suggestions for Authors

The authors should omit any discussion of analyte changes during the microwave process that are not significant.

  • According to the Reviewer’s suggestion, the significant changes in analyzed bioactive compounds content in responce to popping were emphasized and disscused.

The use of "flakes" is not common to describe popped popcorn.  More appropriate is "popcorn" or "kernels"

  • The term "flakes" is rather common to describe popped popcorn, which is well documented by the References no. 12, 17 and 20.

line 53 per- not per

  • In the manuscript main body text, there is no per – not per.

line 113 no standard is mentioned.

  • We thought that mentioning the „standard“ in the title of the Subsection 3.3. Spectrophotometric methods is inadequate and unnecessary.

On each table at the bottom of the data the authors should put the average of all the 12 hybrids and the standard deviation and the statistics for the average % changes due to the microwave process.

  • We thanked the Reviewer for useful suggestion, and we provided the average values for each evaluated parameter, for all tested hybrids. In addition, we thought that it would be more correct to provide average values for % change separately for both the increase and the decrease.

Reviewer 3 Report

Comments and Suggestions for Authors

Refree report_molecules-ID :2819060

The authors studied the effect of microwave popping on antioxidants determined in twelve popcorn hybrids and they used statistical analysis to discuss data. The work is interesting but PCA analysis seem not appropriate for the set of data. comments are presented below. Conclusion and abstract should be reformulated after revision of statistical analysis.

Abstract

Line 12, were evaluated instead of was evaluated

Introduction

Lines 57-61: very long sentence to be reformulated and divided in short sentences

Lines 77-80: rewrite the objectives clearly

Material and methods

Line 153: write appropriately DPPH in equation 1, instead of dpph

Line 177, 2.5, the authors have to add the dendrogram analysis, it is an analysis different from PCA in both case, both analysis should be statistically validated, see comments below in result section.

Indication about replication of experiments and analyses should be indicated in different sections of material and methods.

Results and discussion

Table 2, if the value is too high express it differently for example use mmol/g of  DW sample instead of nmol/g of DW sample for GSH for example. These values should be compared with value available in literature, value seem to be very high for GSH.

Did the authors make any dilution before determination of phenols and DPPH analysis?

Line 340, 3.4.PCA, rewrite title PCA of what kind of data !!

Line 363: Fig 1 seems wrong please check, for the analyzed data the % of variation should be the same in PC1 and PC2 in scores (A) and loading plots (B).

Besides it seems that PCA is not appropriate in this case, to be statistically appropriate the sum of total variation in both axes PC1 and PC2 should be equal or superior to 70%. In this case it is lower or equal to 60%, it means that the analyzed factors are not able to represent the entire variance of the system.

The authors have to check if eigenvalues for PC1 and PC2 if they  are superior to 1, if isn’t the case, PCA is not valid and significant for the presented data, and they have to try another data analysis assays.

Comments on the Quality of English Language

Author Response

Comments and Suggestions for Authors               Refree report_molecules-ID: 2819060

The authors studied the effect of microwave popping on antioxidants determined in twelve popcorn hybrids and they used statistical analysis to discuss data. The work is interesting but PCA analysis seem not appropriate for the set of data. comments are presented below. Conclusion and abstract should be reformulated after revision of statistical analysis.

Abstract

Line 12, were evaluated instead of was evaluated

  • Accepted and corrected.

Introduction

Lines 57-61: very long sentence to be reformulated and divided in short sentences

  • According to the Reviewer’s suggestion, the sentence was reformulated and divided into three shorter sentences.

Lines 77-80: rewrite the objectives clearly

  • According to the Reviewer’s comment, the objectives were clearly rewritten.

Material and methods

Line 153: write appropriately DPPH in equation 1, instead of dpph

  • Corrected.

Line 177, 2.5, the authors have to add the dendrogram analysis, it is an analysis different from PCA in both case, both analysis should be statistically validated, see comments below in result section.

  • According to the Reviewer’s comment, In Material and methods Section (now subsection 3.5. Statistical analysis) was added the applied agglomerative Hierarchical Cluster Analysis (HCA).

Indication about replication of experiments and analyses should be indicated in different sections of material and methods.

  • According to the Reviewer’s comment, experimental replications in analyses were indicated throughout all subsections within the Section Material and methods.

Results and discussion

Table 2, if the value is too high express it differently for example use mmol/g of  DW sample instead of nmol/g of DW sample for GSH for example. These values should be compared with value available in literature, value seem to be very high for GSH.

  • According to the Reviewer comment, the GSH content was changed into the µmol/g DW of sample. The values obtained from this study were compared with the values obtained from the cited study on standard quality maize (Reference no. 7).

Did the authors make any dilution before determination of phenols and DPPH analysis?

  • For determination of total phenolic content, before adding Folin-Ciocalteu reagent, supernatant (0.2 mL) was diluted with 0.5 mL of distilled water. For determination of antioxidant activity, an aliquot of the supernatant (0.3 mL) was diluted with 0.2 mL acetone/water mixture (7:3, v/v), as described in the subsection 3.3.4. within the Section Material and methods.

Line 340, 3.4.PCA, rewrite title PCA of what kind of data !!

  • Corrected and rewritten.

Line 363: Fig 1 seems wrong please check, for the analyzed data the % of variation should be the same in PC1 and PC2 in scores (A) and loading plots (B).

Besides it seems that PCA is not appropriate in this case, to be statistically appropriate the sum of total variation in both axes PC1 and PC2 should be equal or superior to 70%. In this case it is lower or equal to 60%, it means that the analyzed factors are not able to represent the entire variance of the system.

The authors have to check if eigenvalues for PC1 and PC2 if they  are superior to 1, if isn’t the case, PCA is not valid and significant for the presented data, and they have to try another data analysis assays.

  • We thanked the Reviewer for pointing out the mistake when presenting the PCA results. Accordingly, the loading plot was replaced. We also agreed with the Reviewer that the sum of total variation should be equal or superior to 70%, which in our case is 82.43%. As the Reviewer additionally reported, the Eigenvalues for PCs should be superior to 1 for valid and significant PCA. In the case of this study, the first four PCs are with Eigenvalues superior to 1, pointing out to the validity and significance of the presented data. To support our findings, we additionally provided the scores for PC1-PC3 and PC1-PC4 and their complementary loading plots, as well as the Figure 1S with Eigenvalues in Supplementary material.

Reviewer 4 Report

Comments and Suggestions for Authors

The study entitled as “Assessment of the antioxidant status of microwave popped popcorn” has focused on the effect of microwave popping process on the phytochemical content of 20 maize hybrids. For this purpose, the authors have investigated total phenolic content (TPC), glutathione (GSH), phytic P (Pphy), carotenoids (i.e. lutein, zeaxanthin and β-carotene) and tocopherols (i.e. α-, β-+ γ- and δ-) content, as well as the antioxidant activity of maize kernels and flakes.

 There is a similar study that deals with the microwave popping of maize (https://doi.org/10.1016/j.jcs.2016.05.013). The difference of the submitted study is the use of 20 hybrid maize types and more detailed analyses for phytochemicals.

 The title of the study will be revised since the only studied fact is not antioxidant status. The abstract needs to be revised since no numerical results were given. The introduction part of the manuscript states the problematic of the study well. However, the objective part needs revision. Materials and methods part is successful is giving details about the process.

 The first part of the results section (lines: 191 – 199) is not clear to me. The objective of this part and application of ANOVA are not clear.

 In general (sections 3.1, 3.2, 3.3), the Authors preferred to express the difference between the kernels and flakes via %increase. However, a statistical analysis between the kernels and flakes (values in rows) will better express the difference between them.

 In general, discussions on the possible physical/chemical reasons of obtained results are satisfactory.

 However, please see my extra comments in the uploaded pdf under the comments section.

Author Response

Comments and Suggestions for Authors

The study entitled as “Assessment of the antioxidant status of microwave popped popcorn” has focused on the effect of microwave popping process on the phytochemical content of 20 maize hybrids. For this purpose, the authors have investigated total phenolic content (TPC), glutathione (GSH), phytic P (Pphy), carotenoids (i.e. lutein, zeaxanthin and β-carotene) and tocopherols (i.e. α-, β-+ γ- and δ-) content, as well as the antioxidant activity of maize kernels and flakes.

There is a similar study that deals with the microwave popping of maize (https://doi.org/10.1016/j.jcs.2016.05.013). The difference of the submitted study is the use of 20 hybrid maize types and more detailed analyses for phytochemicals.

The title of the study will be revised since the only studied fact is not antioxidant status.

  • The title was revised according to the Reviewer’s comment.

The abstract needs to be revised since no numerical results were given.

  • The abstract was revised and numerical results were provided.

The introduction part of the manuscript states the problematic of the study well. However, the objective part needs revision.

  • The objective part was revised in the Introduction Section.

Materials and methods part is successful is giving details about the process.

The first part of the results section (lines: 191 – 199) is not clear to me. The objective of this part and application of ANOVA are not clear.

  • At the beginning of the Results and Discussion Section, the explanation for ANOVA application was provided.

In general (sections 3.1, 3.2, 3.3), the Authors preferred to express the difference between the kernels and flakes via %increase. However, a statistical analysis between the kernels and flakes (values in rows) will better express the difference between them.

  • The LSD test was done to express the differences between the raw kernels and the flakes regarding the bioactive compounds content. In order to determine the trend of evaluated phytochemicals content in response to popping treatment, a percent change (% change) was used.

In general, discussions on the possible physical/chemical reasons of obtained results are satisfactory.

However, please see my extra comments in the uploaded pdf under the comments section.

No numerical results were given in the abstract.

  • The numerical results were provided in the Abstract.

is it daily?

  • It was meant on daily basis.

these referred studies are dealing with the health-promoting roles of antioxidants, however, this sentence needs a reference giving proof about the presence of antioxidants in maize. please revise.

  • According to the Reviewer’s suggestion, updated references that refer to evaluated phytochemical content observed in maize were provided.

I dont think that the number of investigated antioxidant material increases the novelty of the study.  In fact, the study in ref#9 deals with 5 types of antioxidant content. therefore this isnot the first study. I believe instead of this sentence (78-79) the authors will emphasize the value of this study better.

  • Accepted and corrected.

this is not clear. as mentined in the text: "Popping treatment was performed in microwave oven (Samsung, model MS23F301T) at 700 W for 5 min on Instant program", popping treatment was only applied under one condition. how did the authors evaluate the effect of popping treatment by ANOVa? please kindly explain.

  • The effect of popping was estimated by comparison of values for observed phytochemical compounds between raw kernels and popcorn flakes using two-way ANOVA.

caption of table 1 is not descriptive to a reader who only examine this table.

  • We completely agreed with the Reviewer, and we provided the abbreviations for the ANOVA factors and evaluated bioactive compounds in the Table 1 caption.

please kindly add: the comparison of  values were evaluated between colums or rows, or any other?

  • Accepted and added.

the term "enrichment" is generally used for the situations in which a natural or artificial additive is added to the product or formulation. however, in this part of the study, no additive was added. TPC was increased due to a treatment.

  • Accepted and omitted from the manuscript main body text.

not clear. which compounds? please specify.

  • The compounds were specified.

this discussion is not satisfactory for this kind of dramatic difference btw hybrids.

  • According to the Reviewer’s comment, the text was revised and some parts were omitted.

please kindly add: the comparison of  values were evaluated between colums or rows, or any other?

  • Accepted and added.

please also consider the solubility of carotenoids in oil.

  • According to the Reviewer’s suggestion, the sentence was improved.

sentence fragment, please revise.

  • According to the Reviewer’s suggestion, the sentence was improved.

Round 2

Reviewer 4 Report

Comments and Suggestions for Authors

I have gone through each response of authors. I have also checked the responses embedded in the manuscript. As the Reviewer 4, I want to inform you that the Authors have successfully responded or revised the manuscript according to all my previous comments.